# Sitting Pressure Measurements in Wheelchair Users—Can the Effects of Daily Relief Activities Be Depicted?

**DOI:** 10.3390/s24123806

**Published:** 2024-06-12

**Authors:** Roy Müller, Clara Oette, Cedric Oette, Lucas Schreff, Rainer Abel

**Affiliations:** 1Department of Orthopedic Surgery, Klinikum Bayreuth GmbH, 95445 Bayreuth, Germany; clara.oette@stud.uni-heidelberg.de (C.O.); lucas.schreff@klinikum-bayreuth.de (L.S.); frank-rainer.abel@klinikum-bayreuth.de (R.A.); 2Bayreuth Center of Sport Science, University of Bayreuth, Universitätsstraße 30, 95447 Bayreuth, Germany; 3University Hospital Erlangen, Friedrich-Alexander-University Erlangen, 91054 Erlangen, Germany; 4Hawk Intelligent Technologies GmbH, Schafäckerlein 23, 91413 Neustadt an der Aisch, Germany

**Keywords:** wheelchair, pressure ulcer, pressure mapping, spinal cord injury, quality of life, dispersion index

## Abstract

Seat pressure measurements in wheelchair users have been available for some time; however, repeated measurements from a commercially available pressure mat over 90 min did not differ in the pressure-loaded measurement area or the coordinates of the center of pressure, even in participants who were able to reposition themselves in the wheelchair. The question therefore arises: to what extent are there other parameters that reflect the activity of wheelchair users with the pressure mat? To investigate this, a commercial pressure mat (BodiTrak^®^) was used to perform the measurements of pressure of 33 adult wheelchair-dependent people with spinal cord injury after 30 and 90 min sitting on the cushion. In addition to the standard output of the pressure mat, graph-based surface analyses (calculation of the area of maximum pressure, calculation of the pressure-loaded measurement area, and pressure–area ratio) was performed retrospectively using Python 3.7. The analysis of the measurements after 30 and 90 min was performed by distinguishing the participants between those who could actively change their position (N = 24) and those who could not (N = 9). The parameters of the pressure mat and the graph-based analyses remained unchanged for active participants. In participants who were unable to actively change their position, the area of maximum pressure and the pressure–area ratio (ratio of maximum pressure area and total pressure-loaded area) increased. Significant differences between minutes 30 and 90 are only found for the pressure–area ratio. Thus, when measuring the seat pressure of wheelchair users, the pressure–area ratio should be taken into account as it reflects the daily relief activities of wheelchair users.

## 1. Introduction

Pressure ulcers are one of the most serious complications in people with spinal cord injuries (e.g., [1,2,3]). The development of pressure ulcers is influenced by a variety of extrinsic and intrinsic factors, whereas pressure and load duration have been discussed to be the primary contributors (e.g., [4,5]). In order to reduce the risk of developing a pressure ulcer, the load duration can be influenced by relief activities (e.g., leaning on the side or pushing up [6]), which promote blood circulation [7,8,9].

Pressure distribution within the seating surface can be measured using a seating pressure mat. Seat pressure measurements in wheelchair users have been available for quite some time (e.g., [10,11,12]); however, there has been little research that looked specifically at prolonged sitting on a pressure mat (i.e., sitting for 90 min or longer) and the effects of daily relief activities on the pressure-loaded measurement area. There are few studies that address sitting activities lasting more than a week (e.g., [13,14]); however, these studies use weight shift sensors attached to the support seat of a manual wheelchair rather than a pressure sensing mat. Long-term studies with a pressure measuring mat are best performed on participants’ hospital beds (e.g., [15,16,17]). These studies address the effect of a continuous (at least 72 h) bedside pressure mapping system for reducing interface pressures, but do not directly address the activity of the patients.

Everyday activities, such as moving in the wheelchair or changing the sitting position, are expected to show notable changes in the peak pressure areas and the center of pressure, especially when active changes of position are possible. Contrary to this assumption, the values of the parameters measured using a commercial seat pressure mat (e.g., mean pressure, maximum pressure, pressure-loaded measurement area, and the coordinates of the center of pressure) were very similar after 30 and 90 min sitting on the cushion and moving as desired and in accordance with the necessities of their daily activities [18]. In addition, participants who were able to relieve themselves in the wheelchair as needed and wanted showed almost the same results as participants who were not able to actively change their position [18].

The question arises as to what extent other parameters, which were not recorded by the commercially available seat pressure mat, reflect the influence of daily relief activities. Thus, the aim of this retrospective study was to analyze whether there is a difference in other parameters (e.g., the ratio of high-pressure area and total pressure-loaded area) between people who are able to actively reposition themselves and those who are not.

## 2. Materials and Methods

### 2.1. Participants

Thirty-four participants with spinal cord injuries were recruited as part of the routine examination at the Spinal Cord Injury Unit at the Klinikum Bayreuth GmbH (Hohe Warte), Germany [18]. The sample size was calculated using an a priori power analysis for an ANOVA model conducted by means of G*Power 3.1.5 software, given the following input parameters: effect size: F = 0.4; alpha error probability: 0.05; power: 0.8; number of groups: one [18]. Due to the lack of relevant information about each individual, we analyzed data from 33 participants in this retrospective study. Twenty-four of these participants were able to actively change their position; nine were unable to actively change their position (Table 1).

Exclusion criteria were severe cognitive impairments that precluded participation. Informed written consent was obtained from each volunteer. The investigation was approved by the ethics review board of the Friedrich–Alexander University Erlangen-Nürnberg, Germany (480_18B) and was conducted in accordance with the Declaration of Helsinki.

### 2.2. Measurements

The experimental methods have been presented by Oette et al. [18] and, thus, we will only briefly report the measurement and setup. Measurements were taken using a commercial sitting pressure mat from BodiTrak^®^ (Winnipeg, MB, Canada) (16 × 16 = 256 sensors) and the corresponding software FSA.1, Canada (e.g., [20,21]). At the beginning of the measurement period, the seat pressure mat was placed on the seat cushion and the participants were positioned on the mat in their own wheelchair by a qualified occupational therapist (Figure 1). Participants sat on the pressure mat for the entire 90 min and were not repositioned for the measurements after 30 and 90 min. As no continuous measurement was carried out, the pressure mat was only connected to the PC via a data cable to record one measurement (image) after 30 and 90 min (Figure 1). Before and after both measurement points, the participants were encouraged to move as desired and in accordance with the necessities of their activities. They were also asked to perform pressure relief maneuvers as usual.

### 2.3. Data Processing

Using the standard supplied FSA.1 software package of the commercial seat pressure mat, the following relevant parameters were measured at minutes 30 and 90: maximum pressure (mmHg), mean pressure (mmHg), and measurement area (m^2^) as well as the horizontal and vertical coordinates of the center of pressure (cm). Furthermore, a graph-based surface analyses was performed using Python 3.7. The graph-based analysis tasks were the calculation of the pressure-loaded measurement area (m^2^), the maximum pressure area (m^2^), and the pressure–area ratio (as the ratio of the maximum pressure area to the total pressure-loaded measurement area). Due to sensor saturation, a maximum of only 200 mmHg can be read; therefore, the range of maximum pressure was calculated for the values of 200 mmHg. These maximum pressure areas correspond to the red areas in the measurement images (Figure 1). The shape of the seat pressure area was analyzed in a two-step procedure for the graphical processing of the three-dimensional measurement results. First, the maximum pressures (red areas) were extracted and converted into a binary image (Figure 2). Second, the size of the maximum pressure areas was analyzed by self-written algorithms and the supporting library OpenCV.

### 2.4. Statistical Analyses

Statistical analyses were performed using SPSS 26 (Chicago, IL, USA). To test the normality of distributions, Kolmogorov–Smirnov tests were implemented for each parameter of the pressure mat (i.e., pressure-loaded measurement area) and the graph-based analyses (i.e., pressure-loaded measurement area, area of maximum pressure, and pressure–area ratio). The validity of the pressure-loaded measurement area, between both the outcome of the commercial pressure mat and the graph-based analyses, was assessed by calculating Pearson’s correlation. The analysis of the measurements after 30 and 90 min was conducted for all participants as well as separately for participants who were able to actively change their position (active participants) and for participants who were not able to (inactive participants). To evaluate the parameters of the pressure mat and the graph-based analyses between minutes 30 and 90, we performed a paired *t*-test. An alpha level of 0.05 was used for all statistical tests.

## 3. Results

The parameters of the pressure mat and the graph-based analyses were normally distributed for all participants (N = 33), but also for active (N = 24) and inactive (N = 9) participants. The correlations between the pressure-loaded measurement area between the pressure mat and the graph-based analyses were highly significant for all participants (minute 30: r = 0.99, *p* < 0.001; minute 90: r = 0.99, *p* < 0.001), but also for participants who were able to actively change their position (minute 30: r = 0.99, *p* < 0.001; minute 90: r = 0.99, *p* < 0.001) and for participants who were not able to actively change their position (minute 30: r = 0.99, *p* < 0.001; minute 90: r = 0.99, *p* < 0.001). However, significant differences between minutes 30 and 90 can only be found in the pressure–area ratio of inactive participants (Table 2).

## 4. Discussion

Our research shows that the pressure–area ratio (the ratio of the maximum pressure area to the total pressure-loaded measurement area) can be used in long-term seat pressure measurements to reflect the daily relieving activities of wheelchair users.

More precisely, the parameters of the graph-based analyses (i.e., pressure-loaded measurement area, area of maximum pressure, and pressure–area ratio) remained unchanged for all participants but also for participants who were able to actively change their position (Table 2). This finding is in accordance with the standard output of the commercial seat pressure mat [18]. In participants who were unable to actively change their position, the parameters measured with the commercially available pressure mat from Oette et al. [18] remained unchanged, too, but the graph-based parameters differ. More precisely, the area of maximum pressure increased by about 87% and the pressure–area ratio increased by about 52% from minutes 30 to 90 (Table 2). This means that, without unloading the piezoelectric sensors of the pressure mat and/or without unloading the cushion, the area of maximum pressure and, thus, the pressure–area ratio, increase slowly over time; however, significant differences between minutes 30 and 90 can only be found in the pressure–area ratio (Table 2).

Another potentially interesting parameter, which can be measured by a recent update to the commercial pressure mat (BodiTrak Pro), is the dispersion index. The dispersion index was defined as the sum of pressure distributed over a region (e.g., the ischial tuberosity or the sacrococcygeal region [9,22,23]) divided by the sum of pressure readings over the entire sensor mat, calculated to provide information about the relative pressures under areas at high risk of tissue damage. However, the dispersion index could not be measured by Oette et al. [18], and the effect of pressure relief activities (e.g., leaning to the side or pressing up [6]) as well as their frequency and duration on the dispersion index still need to be investigated.

Some limitations of the present study require consideration. First, due to sensor saturation, no maximum pressure values beyond 200 mmHg could be measured using the commercial BodiTrak pressure mat. This means that the area of maximum pressure might look different if the mat could measure more than 200 mmHg; however, this should not have a major impact when using the 200 mmHg as a threshold value, which serves as the basis for calculating the pressure–area ratio in our study. Second, the pressure of the pressure sensors, which slowly increases over time (sensor creep), influences the measurement results; however, due to the inertia of the pressure measurement sensors, the error can be sufficiently reduced if you wait longer. It was therefore recommended that the first measurement be taken between 6 [9] and 8 min [24] after positioning the participants. However, we assume that this sensor creep is no longer as great with a waiting time of 30 and 90 min [18]. Third, for the duration of the study, participants were encouraged to move as desired and in accordance with the necessities of their daily relief activities. Since we did not record the specific activities of the participants during the 90 min, the effect of the relief activities cannot be assessed. This should be taken into account in future studies.

## 5. Conclusions

The parameters of both the outcomes of the commercial pressure mat and the graph-based analysis remained unchanged for active participants; however, in participants who were unable to actively change their position, the pressure–area ratio increased. Thus, when measuring the seat pressure of wheelchair users, the pressure–area ratio should be taken into account as it reflects the daily relief activities of wheelchair users.

## Figures and Tables

**Figure 1 sensors-24-03806-f001:**
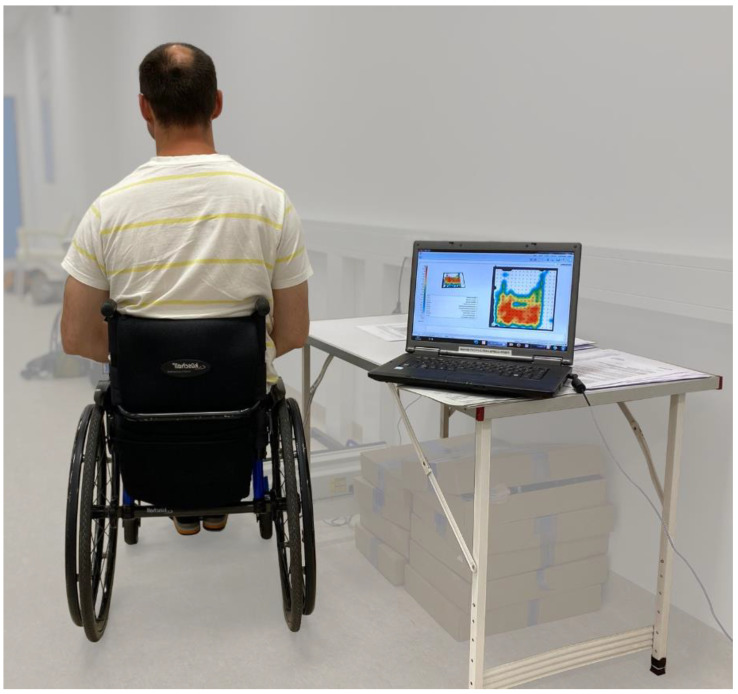
Example of a seat pressure measurement. Measurements were taken using a pressure mat (BodiTrak^®^) placed on the participants’ own seat cushion in their own wheelchair.

**Figure 2 sensors-24-03806-f002:**
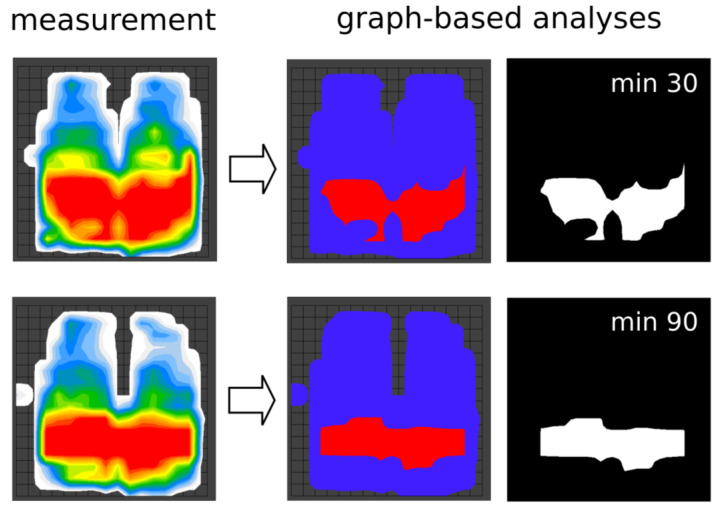
Example of a seat pressure measurement for both the outcome of the commercial pressure mat (measurement) and the corresponding graph-based analysis at minutes 30 and 90. For the graph-based analysis, the maximum pressures (red areas) were extracted and converted into a binary image.

**Table 1 sensors-24-03806-t001:** Participants’ characteristics separately for participants who were able to actively change their position (active participants) and for participants who were not able to actively change their position (inactive participants).

	Active Participants (N = 24)	Inactive Participants (N = 9)
weight	73.4 ± 17.2	78.0 ± 24.5
size	173.2 ± 12.0	177.4 ± 8.5
age	45.3 ± 18.4	54.2 ± 12.5
sex	20 male/4 female	7 male/2 female
cervical spine lesion	NA	8 incomplete/1 complete
thoracic spine lesion	10 incomplete/12 complete	NA
lumbar spine lesion	1 incomplete/1 complete	NA
ASIA Scale ^1^	13 A/1 B/6 C/4 D	1 A/2 B/4 C/2 D
decubitus ulcer in the past	15 no/9 yes	3 no/6 yes

^1^ ASIA: American Spinal Injury Association Impairment Scale (e.g., [19]).

**Table 2 sensors-24-03806-t002:** Mean ± SD of the seat pressure measurements of both the outcome of the commercial pressure mat and the graph-based analyses.

	Parameter of the Pressure Mat		
		min 30	min 90
pressure-loaded measurement area (m^2^)	all participants	0.129 ± 0.029	0.129 ± 0.030
	active participants	0.130 ± 0.030	0.128 ± 0.031
	inactive participants	0.126 ± 0.027	0.132 ± 0.028
	**Parameter of graph-based analyses**		
		**min 30**	**min 90**
pressure-loaded measurement area (m^2^)	all participants	0.161 ± 0.031	0.161 ± 0.031
	active participants	0.162 ± 0.033	0.160 ± 0.034
	inactive participants	0.158 ± 0.026	0.163 ± 0.026
area of maximum pressure (m^2^)	all participants	0.021 ± 0.014	0.023 ± 0.017
	active participants	0.023 ± 0.015	0.021 ± 0.017
	inactive participants	0.015 ± 0.011	0.028 ± 0.018
pressure–area ratio	all participants	13.31 ± 8.36	13.89 ± 9.71
	active participants	13.92 ± 8.95	12.44 ± 9.22
	inactive participants	11.68 ± 6.75	17.77 ± 10.47 *

* Significant differences between measurement at minutes 30 (min 30) and 90 (min 90) are indicated with ‘*’ (*p* < 0.05).

## Data Availability

The datasets generated and analyzed during the current study are available on request from the corresponding author, R.M.

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
