# Peer review of "Sitting Pressure Measurements in Wheelchair Users—Can the Effects of Daily Relief Activities Be Depicted?"

_sensors, 2024, doi:10.3390/s24123806_

Round 1
Reviewer 1 Report
Comments and Suggestions for Authors
What is the novelty of this work? Kindly highlight it in the introduction section.
The aim of the study is unclear. Kindly clearly mention in the abstract and introduction section.
Kindly add more literature with the gap in the introduction section regarding the pressure distribution and its effect for wheelchair users.
How did the authors calculate the sample size?
How many trials were conducted to collect the data while seating for 30 mins and 90 mins.
kindly explain table 2 in the results section rather than discussion section.
If possible, any data available from normal wheelchair users? so the data can be compared between two groups.
In discussion first paragraph kindly add study prominent outcome.
Kindly add more to the discussion section
Comments on the Quality of English Languageneed to check after final version is approved
Author Response
Dear reviewer,
Thank you very much for your valuable and helpful comments that guided us in improving the manuscript.
Changed manuscript text is marked in blue in both the manuscript and our responses to your comments.
Reviewer #1
What is the novelty of this work? Kindly highlight it in the introduction section.
Response: Thank you for this comment. We revised the introduction section to make this more clearly.
The aim of the study is unclear. Kindly clearly mention in the abstract and introduction section.
Response: Yes, you are right. We revised the last paragraph of the introduction section. It now reads: “The question arises to what extent other parameters that were not recorded by the commercially available seat pressure mat reflect the influence of daily relief activities. Thus, the aim of this retrospective study was to analyze whether there is a difference in other parameters (e.g., the ratio of high-pressure area and total pressure-loaded area) be-tween people who are able to actively reposition themselves or not.”
Kindly add more literature with the gap in the introduction section regarding the pressure distribution and its effect for wheelchair users.
Response: Done as suggested.
How did the authors calculate the sample size?
Response: The sample size was calculated using an a priori power analysis for an ANOVA model conducted by means of G*Power 3.1.5 software, given the following input parameters: effect size F = 0.4 (detectable), alpha error probability: 0.05, power: 0.8, and number of groups: 1 [13]. We added this information in the method section.
How many trials were conducted to collect the data while seating for 30 mins and 90 mins.
Response: After 30 minutes of sitting, a single measurement was taken, and a second one 60 minutes later. To make this more clearly, the following sentences were added to the methods section: “At the beginning of the measurement period, the seat pressure mat was placed on the seat cushion and the participants were positioned on the mat in their own wheelchair by a qualified occupational therapist (Figure 1). Participants sat on the pressure mat for the entire 90 minutes and were not repositioned for the measurements after 30 and 90 minutes. As no continuous measurement was carried out, the pressure mat was only connected to the PC via a data cable to record one measurement (image) after 30 and 90 minutes (Figure 1).”
kindly explain table 2 in the results section rather than discussion section.
Response: Done as suggested.
If possible, any data available from normal wheelchair users? so the data can be compared between two groups.
Response: What does the reviewer understand by “normal wheelchair users”? Subjects without spinal cord injury? However, we have no further data from other groups of people that we can include in our analysis.
In discussion first paragraph kindly add study prominent outcome.
Response: Good idea. Thank you. We added the following first paragraph to the discussion: “Our research shows that the pressure area ratio (the ratio of the maximum pressure area to the total pressure-loaded measurement area) can be used in long-term seat pressure measurements to reflect the daily relieving activities of wheelchair users.”
Kindly add more to the discussion section
Response: Done as suggested. See discussion section.

Reviewer 2 Report
Comments and Suggestions for Authors
Abstract – no specific feedback about the actual abstract, although changes might be needed based on feedback from the remainder of the article, if the authors make the recommended changes.
Overall concerns:
There is no discussion of the known issues with seat interface pressure mapping systems around “sensor creep” – which causes the recorded interface pressures to gradually increase over time and “sensor saturation” – which is what is happening when the sensors read 200 mmHg, therefore cannot read any higher pressures.
The use of the term “dispersion index” is not the same as it has been defined in previous work – including the two works cited. The classic definition of “dispersion index” is the total pressure recorded in a defined area of the sensory array (typically the area under the ischial tuberosities and sacral region of the seated person), divided by the pressures under the whole mat. It provides information about the relative pressures under these high risk areas of the person. The authors need to include information regarding these differences in the use of this term.
More information needed about the exact data collection. I know they cite another study, but I think there is a few bit more information needed about how the data were collected to make it clear to the reader.
Questions:
1. I am assuming the participants sat on the pressure mats for the entire 90 minutes?
2. How long were the data collected? If collected over the duration of sitting (i.e. the full 90 minutes), what was the interval used for recording the IPM data eg. One reading every minute or every 2 minutes? Was it just one image collected at minutes 30 and 90? A little more detail would be very helpful for the reader to understand how the data were collected.
3. I assume the data were collected wirelessly (so the wired attachment to a data collection device did not interfere with their ability to move around, please confirm if this is the case.
Specific questions/concerns:
Abstract – see concerns in the sections below. I don’t have any specific concerns about the abstract, but would like to see information included indicating a discussion of the known concerns of “sensor creep” , “sensor saturation” , and the different use of the term “dispersion index” in this research study (with more detailed explanation in the full text of the sections - see below).
Introduction
Line 44 – 45: the sentence indicating “little research into the stability of measurements over time…” is not really very accurate, although there certainly may be little research that looked specifically at prolonged sitting – i.e. sitting of 90 minutes (or longer). I believe a more thorough literature search is needed to capture the longer duration studies that have been done.
For example:
Ho, Chester, et al. "Effect of a Continuous Bedside Pressure Mapping System for Reducing Interface Pressures: A Randomized Clinical Trial." JAMA Network Open 6.6 (2023): e2316480-e2316480.
Line 54 – “relief” should be “relieve”
Lines 56-57 – I think this is a flawed assumption and not founded on the information provided. There are many possible reasons why pressure mapping parameters don’t accurately capture these changes – many of them are related to the limitations of the sensing technologies, so I don’t think this summary statement is well founded.
2. Materials and methods
2.2 Measurements – I realize there is a reference to an earlier study that includes a full description of the data collection process, however I think there are key factors that need to be included in this description:
1. Was the mat positioned under the patient, then left in place for the full 90 minutes?
2. were the data recorded continuously during the 90 minutes? Or was the capturing of data just performed at certain time intervals, if so what were the intervals, how many “frames” of data were collected, what interval was the data collection.
3. were the data collected wirelessly? (I assume so, but this should be specified)
2.3 data processing
Using just one time measures? (i.e. a single frame of data from the FSA software);
Line 100: Dispersion index definition is not the standard definition used in the industry (nor in the citations provided), also not what is calculated in the FSA software.
Line 101 – correctly identifies 200 mmHg as the maximum pressure that the system can record, need to explain this as “sensor saturation” – more accurate to report these as saturated sensors, since there is no way to know what the actual pressure is once this value is reached.
Line 144 -149 – there are more parameters that can be obtained from the FSA software, so this section is mischaracterizing what is (or is not), available. Again, the reference to dispersion index as created in the system is inaccurate.
Line 156 (and following paragraph) – there are many additional limitations of this study that have not been mentioned and again, there is no real discussion of some the known limitations of pressure mapping technologies used for this “long duration” testing – when the issues of sensor saturation, creep, and other phenomena become much more impactful.
Conclusions section
Line 168 – again, use of the term “dispersion index” is problematic. The authors should create another term as this terminology in the current application will not be consistent with the usual use of this term.
Line 171 – yes, the dispersion index is available in the FSA software, but it is not the same as what is proposed in this manuscript.
References,
I don’t believe that the 18 references cited in this study are adequate to support the foundations or interpretations of this study. In particular, the authors have misrepresented what is commonly meant by “dispersion index” in the literature (even in the literature cited).
Comments on the Quality of English LanguageOverall the quality of the English Language is very good, with just a few improper word forms detected. I do think there may have been some issues with translation in some of the technical aspects of pressure mapping, which might have made these sections seem inaccurately presented. I am not sure about this.
Author Response
Dear reviewer,
Thank you very much for your valuable and helpful comments that guided us in improving the manuscript.
Changed manuscript text is marked in blue in both the manuscript and our responses to your comments.
Reviewer #2
There is no discussion of the known issues with seat interface pressure mapping systems around “sensor creep” – which causes the recorded interface pressures to gradually increase over time and “sensor saturation” – which is what is happening when the sensors read 200 mmHg, therefore cannot read any higher pressures.
Response: Yes, you are right. We added a discussion of the known issues with seat interface pressure mapping system: “Some limitations of the present study require consideration. First, due to sensor saturation no maximum pressure values beyond 200 mmHg could be measured using the commercial BodiTrak pressure mat. That means that the area of maximum pressure might look different if the mat could measure more than 200 mmHg. However, this should not have a major impact when using the 200 mmHg as a threshold value, which serves as the basis for calculating the pressure area ratio in our study. Secondly, the pressure of the pressure sensors, which slowly increases over time (sensor creep), influences the measurement results. However, due to the inertia of the pressure measurement sensors, the error can be sufficiently reduced if you wait longer. Therefore, it was recommended that the first measurement be taken between six [9] and eight minutes [24] after positioning the participants. However, we assume that this sensor creep is no longer as great with a waiting time of 30 and 90 minutes [18].”
The use of the term “dispersion index” is not the same as it has been defined in previous work – including the two works cited. The classic definition of “dispersion index” is the total pressure recorded in a defined area of the sensory array (typically the area under the ischial tuberosities and sacral region of the seated person), divided by the pressures under the whole mat. It provides information about the relative pressures under these high risk areas of the person. The authors need to include information regarding these differences in the use of this term.
Response: Thank you for this valuable comment. You are right, our dispersion index definition differs from that used in previous works. Therefore, we replaced the term “dispersion index” by the term “pressure area ratio”.
More information needed about the exact data collection. I know they cite another study, but I think there is a few bit more information needed about how the data were collected to make it clear to the reader.
Questions:
1. I am assuming the participants sat on the pressure mats for the entire 90 minutes?
Response: Thank you for this comment. To make this more clearly, the following sentences were added to the methods section: “At the beginning of the measurement period, the seat pressure mat was placed on the seat cushion and the participants were positioned on the mat in their own wheelchair by a qualified occupational therapist (Figure 1). Participants sat on the pressure mat for the entire 90 minutes and were not repositioned for the measurements after 30 and 90 minutes. As no continuous measurement was carried out, the pressure mat was only connected to the PC via a data cable to record one measurement (image) after 30 and 90 minutes (Figure 1).”
2. How long were the data collected? If collected over the duration of sitting (i.e. the full 90 minutes), what was the interval used for recording the IPM data eg. One reading every minute or every 2 minutes? Was it just one image collected at minutes 30 and 90? A little more detail would be very helpful for the reader to understand how the data were collected.
Response: See comment above.
3. I assume the data were collected wirelessly (so the wired attachment to a data collection device did not interfere with their ability to move around, please confirm if this is the case.
Response: See comment above.
Specific questions/concerns:
Abstract – see concerns in the sections below. I don’t have any specific concerns about the abstract, but would like to see information included indicating a discussion of the known concerns of “sensor creep”, “sensor saturation”, and the different use of the term “dispersion index” in this research study (with more detailed explanation in the full text of the sections - see below).
Response: We revised the abstract and replaced the term “dispersion index” by the term “pressure area ratio”.
Introduction
Line 44 – 45: the sentence indicating “little research into the stability of measurements over time…” is not really very accurate, although there certainly may be little research that looked specifically at prolonged sitting – i.e. sitting of 90 minutes (or longer). I believe a more thorough literature search is needed to capture the longer duration studies that have been done.
For example:
Ho, Chester, et al. "Effect of a Continuous Bedside Pressure Mapping System for Reducing Interface Pressures: A Randomized Clinical Trial." JAMA Network Open 6.6 (2023): e2316480-e2316480.
Response: Thank you for this comment. We revised the introduction section to make this more clearly and added more relevant references.
Line 54 – “relief” should be “relieve”
Response: Done as suggested.
Lines 56-57 – I think this is a flawed assumption and not founded on the information provided. There are many possible reasons why pressure mapping parameters don’t accurately capture these changes – many of them are related to the limitations of the sensing technologies, so I don’t think this summary statement is well founded.
Response: We deleted this misleading sentence.
2. Materials and methods
2.2 Measurements – I realize there is a reference to an earlier study that includes a full description of the data collection process, however I think there are key factors that need to be included in this description:
1. Was the mat positioned under the patient, then left in place for the full 90 minutes?
2. Were the data recorded continuously during the 90 minutes? Or was the capturing of data just performed at certain time intervals, if so what were the intervals, how many “frames” of data were collected, what interval was the data collection.
3. Were the data collected wirelessly? (I assume so, but this should be specified)
Response: See comments above.
2.3 data processing
Using just one time measures? (i.e. a single frame of data from the FSA software);
Response: Yes, you are right. To make this more clearly, we added this information. It now reads: “Using the standard supplied FSA.1 software package of the commercial seat pressure mat, the following relevant parameters were measured at minute 30 and minute 90…”
Line 100: Dispersion index definition is not the standard definition used in the industry (nor in the citations provided), also not what is calculated in the FSA software.
Response: See comment above.
Line 101 – correctly identifies 200 mmHg as the maximum pressure that the system can record, need to explain this as “sensor saturation” – more accurate to report these as saturated sensors, since there is no way to know what the actual pressure is once this value is reached.
Response: Yes, you are right. We rephrased the sentence to: ”Due to sensor saturation, a maximum of only 200 mmHg can be read, therefore the range of maximum pressure was calculated for the values of 200 mmHg.”
Line 144 -149 – there are more parameters that can be obtained from the FSA software, so this section is mischaracterizing what is (or is not), available. Again, the reference to dispersion index as created in the system is inaccurate.
Response: You are right. We rephrased the sentence. It now reads: “In participants who were unable to actively change their position, the parameters measured with the commercially available pressure mat from Oette et al. [18] remained unchanged too, but the graph-based parameters differ.”
Line 156 (and following paragraph) – there are many additional limitations of this study that have not been mentioned and again, there is no real discussion of some the known limitations of pressure mapping technologies used for this “long duration” testing – when the issues of sensor saturation, creep, and other phenomena become much more impactful.
Response: Thank you for this comment. We revised the limitation section. It now reads: “Some limitations of the present study require consideration. First, due to sensor saturation no maximum pressure values beyond 200 mmHg could be measured using the commercial BodiTrak pressure mat. That means that the area of maximum pressure might look different if the mat could measure more than 200 mmHg. However, this should not have a major impact when using the 200 mmHg as a threshold value, which serves as the basis for calculating the pressure area ratio in our study. Secondly, the pressure of the pressure sensors, which slowly increases over time (sensor creep), influences the measurement results. However, due to the inertia of the pressure measurement sensors, the error can be sufficiently reduced if you wait longer. Therefore, it was recommended that the first measurement be taken between six [9] and eight minutes [24] after positioning the participants. However, we assume that this sensor creep is no longer as great with a waiting time of 30 and 90 minutes [18]. Third,…”
Conclusions section
Line 168 – again, use of the term “dispersion index” is problematic. The authors should create another term as this terminology in the current application will not be consistent with the usual use of this term.
Response: See comments above. We replaced the term “dispersion index” by the term “pressure area ratio”.
Line 171 – yes, the dispersion index is available in the FSA software, but it is not the same as what is proposed in this manuscript.
Response: See comments above. We replaced the term “dispersion index” by the term “pressure area ratio”.
References,
I don’t believe that the 18 references cited in this study are adequate to support the foundations or interpretations of this study. In particular, the authors have misrepresented what is commonly meant by “dispersion index” in the literature (even in the literature cited).
Response: We added more relevant references and revised the paragraph regarding “dispersion index”.

Round 2
Reviewer 1 Report
Comments and Suggestions for Authors
Thank you for the adjustments. Great work, hope to see more clinical studies from your lab.
Comments on the Quality of English LanguageCan your team check the overall quality.
Reviewer 2 Report
Comments and Suggestions for Authors
greatly improved based on my earlier comments - well done. The reference to dispersion index in the abstract is still a bit problematic, but an accurate description of this concept is provided in the manuscript text that clears up any confusion.